# RNA modification landscape of the human mitochondrial tRNA^Lys regulates protein synthesis

Uwe Richter[1], Molly E. Evans[2], Wesley C. Clark [2], Paula Marttinen[1], Eric A. Shoubridge[3], Anu Suomalainen [4], Anna Wredenberg[5,6], Anna Wedell[6], Tao Pan[2] & Brendan J. Battersby [1]

Post-transcriptional RNA modifications play a critical role in the pathogenesis of human mitochondrial disorders, but the mechanisms by which specific modifications affect mitochondrial protein synthesis remain poorly understood. Here we used a quantitative RNA sequencing approach to investigate, at nucleotide resolution, the stoichiometry and methyl modifications of the entire mitochondrial tRNA pool, and establish the relevance to human disease. We discovered that a $N^1$-methyladenosine (m$^1$A) modification is missing at position 58 in the mitochondrial tRNA^Lys of patients with the mitochondrial DNA mutation m.8344 A > G associated with MERRF (myoclonus epilepsy, ragged-red fibers). By restoring the modification on the mitochondrial tRNA^Lys, we demonstrated the importance of the m$^1$A58 to translation elongation and the stability of selected nascent chains. Our data indicates regulation of post-transcriptional modifications on mitochondrial tRNAs is finely tuned for the control of mitochondrial gene expression. Collectively, our findings provide novel insight into the regulation of mitochondrial tRNAs and reveal greater complexity to the molecular pathogenesis of MERRF.

[1] Institute of Biotechnology, University of Helsinki, Helsinki 00014, Finland. [2] Department of Biochemistry and Molecular Biology, University of Chicago, Chicago, IL 60637, USA. [3] Department of Human Genetics, Montreal Neurological Institute, McGill University, Montreal, QC, Canada. [4] Research Programs Unit - Molecular Neurology, University of Helsinki, Helsinki 00014, Finland. [5] Department of Medical Biochemistry and Biophysics, Karolinska Institutet, Stockholm, Sweden. [6] Centre for Inherited Metabolic Diseases, Karolinska University Hospital, Stockholm 17176, Sweden. These authors contributed equally: Uwe Richter, Molly E. Evans, Wesley C. Clark. Correspondence and requests for materials should be addressed to T.P. (email: taopan@uchicago.edu) or to B.J.B. (email: brendan.battersby@helsinki.fi)

Defects in mitochondrial gene expression are among the most common causes of human mitochondrial disorders[1]. Mitochondria contain a unique set of ribosomes and tRNAs dedicated to the synthesis of the 13 proteins encoded in mitochondrial DNA (mtDNA) (Fig. 1a), all of which are essential for aerobic energy metabolism. Mutations in mitochondrially encoded tRNA genes are the most frequent pathogenic disruptions of mitochondrial protein synthesis[2]. These mutations manifest as diseases with tremendous clinical heterogeneity, in which the tissue specificity, severity, and age of onset do not strictly correlate with the magnitude of the biochemical defect in energy metabolism. Despite the clinical importance of these disorders, the molecular mechanisms by which mitochondrial tRNA mutations affect protein synthesis remain poorly understood.

One such mtDNA mutation is m.8344 A > G in tRNA$^{Lys}$, first identified >25 years ago in patients with MERRF (myoclonus epilepsy, ragged-red fibers) syndrome[3] (Fig. 1b). This mutation causes a severe translation defect during mitochondrial protein synthesis, which has been attributed to a decrease in the steady-state abundance of the mitochondrial tRNA$^{Lys}$, reduced aminoacylation of tRNA$^{Lys}$, and lack of a post-transcriptional 5-taurinomethyl 2-thiouridine RNA modification of the anticodon wobble base[4–7]. A deficiency in the abundance of aminoacylated tRNA$^{Lys}$ would lead ribosomes to stall when the lysine codons AAA or AAG enter the A-site, and chronic stalling would be expected to trigger premature termination of nascent chain synthesis. The 5-taurinomethyl 2-thiouridine modification of U34 prevents superwobbling by restricting decoding of the third codon position to purines[8,9]. In the absence of this modification on tRNA$^{Lys}$ two outcomes are possible: asparagine codons (AAC and AAU) could be misread by tRNA$^{Lys}$, leading to amino acid misincorporation, or altered stability of the codon recognition complex leading to ribosome pausing[10–12]. Individually or collectively, these molecular mechanisms would impinge on mitochondrial protein synthesis.

Mitochondrial protein synthesis can be assayed by metabolic labeling with $^{35}$S methionine in the presence of translation inhibitors of cytoplasmic ribosomes such as anisomycin, so that the radiolabel is only incorporated into mitochondrial nascent chains. Metabolic labeling of mitochondrial protein synthesis in cultured cells homoplasmic for mitochondrial DNA with the m.8344 A > G in tRNA$^{Lys}$ demonstrated the rate of $^{35}$S incorporation is severely impaired, generating specific aberrantly sized polypeptides (e.g., pMERRF) as well as full-length polypeptides all of which are unstable[4,13,14] (Fig. 1c). This pattern of mitochondrial nascent chain synthesis is not consistent with random premature termination at lysine codons[15]. Nonetheless, tRNA stoichiometry and RNA modifications are critical regulators of protein synthesis[16], but the molecular basis by which mitochondrial gene expression is disrupted with this pathogenic tRNA$^{Lys}$ mutation associated with MERRF remains unclear.

To bridge this gap, we capitalized on a methodological advance in next-generation RNA sequencing to investigate the consequences of the MERRF tRNA$^{Lys}$ mutation on the stoichiometry and modifications of the entire mitochondrial tRNA pool in affected patient cells and tissues. Our results implicate a novel tRNA modification in the pathogenesis of MERRF and reveal an extensive and unexpected influence of RNA modifications on the regulation and fidelity of mitochondrial gene expression.

## Results

### Cell type-specific regulation of mitochondrial tRNA abundance.
To investigate transcript-wide regulation of mitochondrial tRNA stoichiometry and RNA modifications in cells harboring the MERRF mutation, we used demethylase-thermostable group II intron RT tRNA sequencing (DM-tRNA-seq)[17]. This method uses a highly processive thermostable group II intron reverse-transcriptase (TGIRT) coupled with two demethylases, to overcome the stable secondary structure and Watson–Crick face methylations of tRNAs to obtain quantitative genome-wide information on cellular tRNA populations and their modifications. We prepared tRNA libraries from deacylated RNA isolated from wild-type myoblasts and myoblasts homoplasmic for the MERRF mutation, with or without demethylase treatment. The demethylase-treated samples generated longer reads, which greatly facilitated specific tRNA assignment and quantitation, whereas the untreated samples allowed identification and quantitative comparison of many modifications at the Watson–Crick face[18]. These libraries were sequenced, and tRNAs were mapped to the human nuclear and mitochondrial genomes (16–40 million total reads per sample, 6–13 million uniquely mapped per sample). Between 12.1–13.4% of all mapped tRNA reads corresponded to mitochondrially encoded tRNAs. The data were highly reproducible among three replicate experiments for both wild-type and homoplasmic mutant cells (Pearson's coefficient > 0.992 for all pairs of replicates; Supplementary Figure 1A). The abundance of mitochondrial tRNA$^{Lys}$ was significantly reduced in MERRF cells (Fig. 1d, e), whereas the steady-state levels of the other 21 mitochondrial tRNAs were essentially identical between wild-type and MERRF cells (Fig. 1e).

The mode of mtDNA transcription can vary across human cell types (so far not studied in myoblasts) and appears to be biased towards nascent RNA synthesis from the promoter of the light coding strand of the genome (Fig. 1a)[19]. We compared the steady-state abundance of each tRNA with the gene position along the mitochondrial coding strands (heavy and light (Fig. 1a)), but observed no trends for either strand (Fig. 1e). Within the mitochondrial genome, the genes encoding tRNA$^{Phe}$ and tRNA$^{Val}$ flank the 12 S rRNA gene (Fig. 1a), and are transcribed together, along with the 16 S rRNA, as a short polycistronic RNA at a level 1.4–8.8-fold greater than genes downstream along the H-strand depending upon the cell type[19]. In HeLa cells, the steady-state levels of tRNA$^{Phe}$ and tRNA$^{Val}$ were 2–3-fold higher compared to those of other H-strand tRNAs when determined by quantitative northern blotting[20] and concordant with nascent RNA synthesis of these genes for this cell type. This difference was not observed in myoblasts using DM-tRNA-seq. To test whether the discrepancy in the steady-state tRNA abundance may be attributable to methodological or actual biological differences in tRNA regulation between diploid and aneuploid cells, we analyzed mitochondrial tRNA stoichiometry in a high-throughput tRNA sequencing dataset obtained from HEK293T cells using DM-tRNA-seq[17]. This analysis revealed that, as in HeLa cells, the levels of tRNA$^{Phe}$ and tRNA$^{Val}$ in HEK293T were 2–3-fold higher than those of other H-strand tRNAs (Fig. 1f, g). Taken together, our data suggest differential regulation of the mitochondrial genome in proliferating human cultured cells, highlighting the need to use diploid cells to investigate the pathogenic mechanisms of human mitochondrial gene expression disorders.

### The m$^1$A58 modification in tRNA$^{Lys}$ is absent in myoblasts with the MERRF mutation.
Post-transcriptional modifications of mitochondrial tRNAs are functionally important to achieve and stabilize the cloverleaf structure, and for decoding during translation elongation[21]. In MERRF, the 5-taurinomethyl 2-thiouridine modification on the anticodon wobble base is missing on the tRNA$^{Lys}$ purified from cybrid cells[7] and from a patient liver biopsy[22]. In the absence of this U34 modification, translation on cognate codons is impaired using an in vitro[5] assay. However,

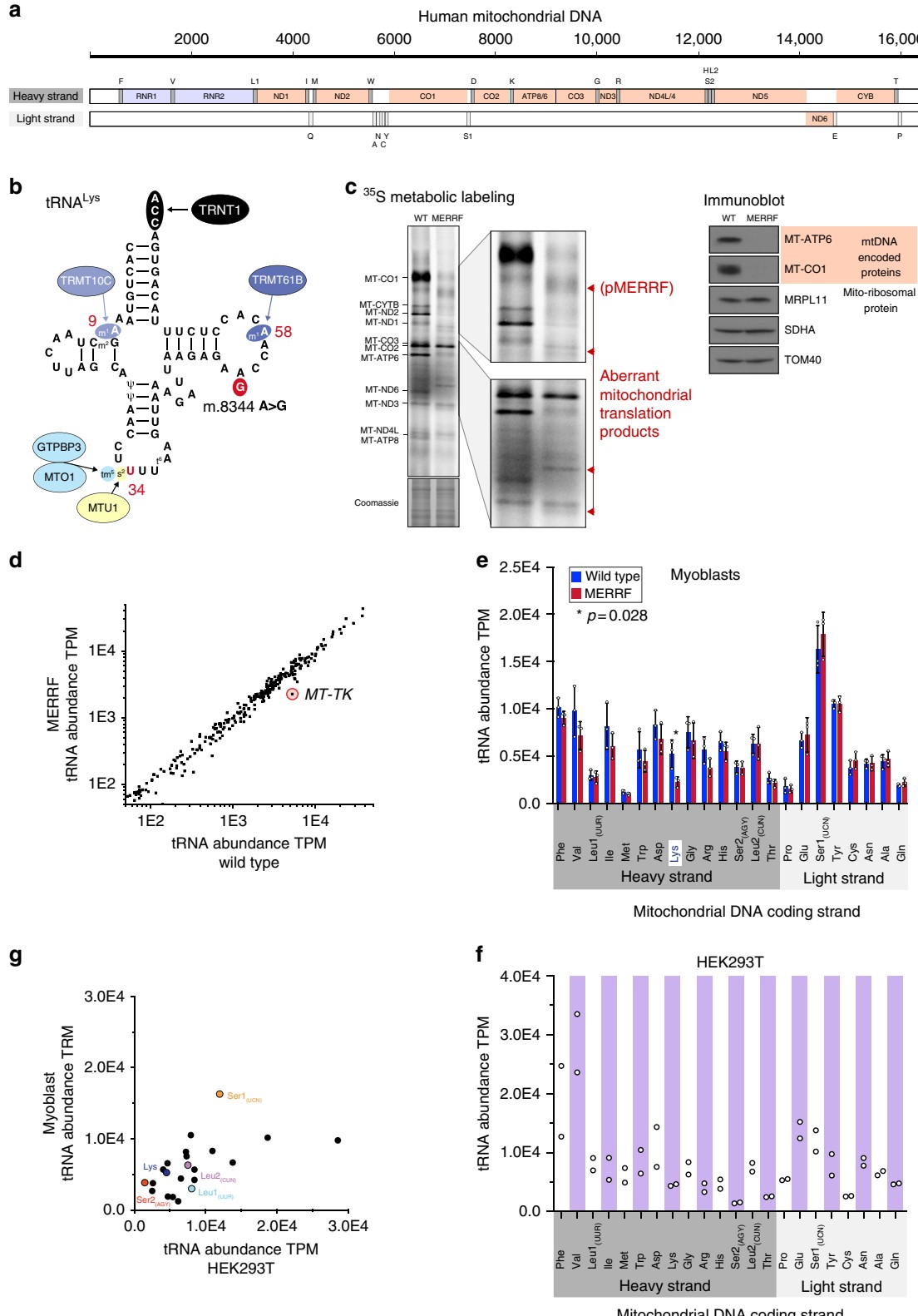

**Fig. 1** Cell type-specific regulation of mitochondrial tRNA stoichiometry. **a** Schematic of mitochondrial DNA illustrating the coding strands. **b** Schematic of mitochondrial tRNA[Lys] highlighting specific RNA modifications and the responsible enzymes. **c** Left, [35]S metabolic labeling into mitochondrial protein synthesis in human myoblasts. Right, immunoblotting of whole-cell lysates. **d** Uniquely aligned reads of mitochondrial tRNAs from human myoblasts from three biological replicates (Pearson's coefficient > 0.992 for all replicate pairs). *MT-TK* indicates the mitochondrially encoded tRNA[Lys]. **e** Mitochondrial tRNA abundance in human myoblasts from (**c**), mean + / − S.D. (*n* = 3). Paired sample *t* test. **f** Mitochondrial tRNA abundance in HEK293T cells from two independent samples. Data are from[17]. **g** Comparison of mean mitochondrial tRNA abundance in wild-type myoblasts vs. HEK293T cells ($r^2 = 0.314$)

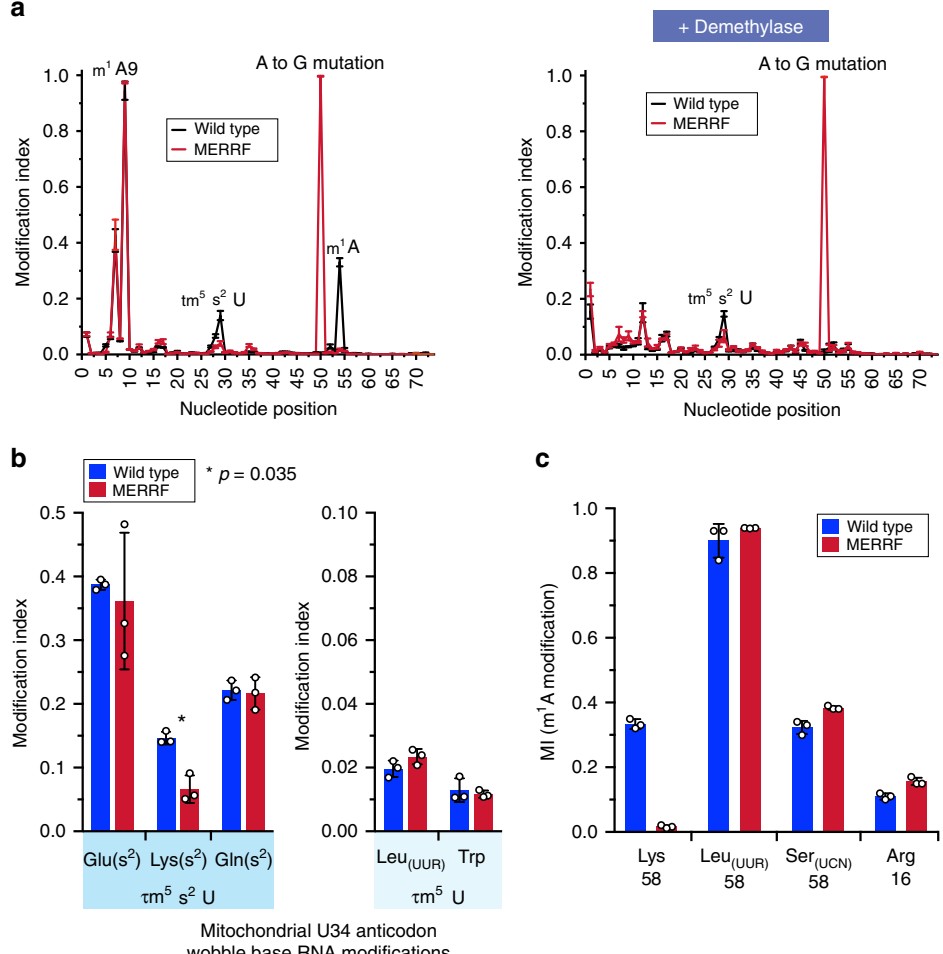

**Fig. 2** Absence of an m$^1$A modification in the m.8344 A > G tRNA$^{Lys}$. **a** Position sequencing plots of tRNA$^{Lys}$ from human myoblasts homoplasmic for wild-type or the m.8344 A > G MERRF mutation without (left) and with (right) demethylase treatment. M$^1$A9 produces two peaks, the first corresponds to RT stop and mutations upon copying the m$^1$A nucleotide and the second stop was due to m$^1$A9 present in the RT template. **b** Quantitative comparisons of U34 anticodon wobble base modifications in mitochondrial tRNAs from human myoblasts. Data represent mean $+/-$ S.D. ($n = 3$). Paired sample $t$ test. **c** Quantification of m$^1$A modifications at the indicated positions in mitochondrial tRNAs from human myoblasts. Data represent mean $+/-$ S.D. ($n = 3$)

metabolic labeling of mitochondrial protein synthesis in human cells is at odds (Fig. 1c) with the in vitro data. Furthermore, lack of the analogous 5-methoxycarbonylmethyl-2-thiouridine (mcm$^5$s$^2$U) U34 modification of the yeast cytoplasmic tRNA$^{Lys}$ does not prevent decoding but rather substantially slows down the rate of translation elongation due to ribosome pausing on lysine codons[10,11]. Therefore, we asked whether other RNA modifications were important to the molecular pathogenesis either directly on tRNA$^{Lys}$ or to the entire pool of mitochondrial tRNAs as part of regulatory feedback from impaired protein synthesis.

DM-tRNA-seq allowed us to assess quantitatively differences in specific RNA modifications across all mitochondrial tRNAs between wild-type and MERRF-mutant cells. At Watson–Crick face methylation sites, such as $N^1$-methyladenosine (m$^1$A), the thermophilic RT generates both stop and read-through mutation sequencing reads. We used the term "modification index" (MI) to quantify the mutation and stop fractions at each nucleotide position[18]. (Please see the Methods section for a discussion comparing the sensitivity and accuracy of DM-tRNA-seq vs. traditional mass-spectrometry approaches for detecting RNA modifications).

First, we addressed the abundance of the τm$^5$s$^2$U anticodon wobble base modification of tRNA$^{Lys}$ between wild-type and

MERRF. As expected, we detected a significant decrease in the modification at this position in MERRF samples (Fig. 2a, b), but it was not completely absent. Unlike $N^1$-methyladenosine (m$^1$A), for which the MI value can be used to approximate the modified fraction, the MI of 0.2 for τm$^5$s$^2$U in wild-type cells likely under-represents the modified fraction because the Watson–Crick face perturbation of τm$^5$s$^2$U is 2-thio, which weakens, but does not block, base pairing between the template and dNTP during cDNA synthesis. Nevertheless, the markedly reduced MI in the MERRF sample was consistent with a reduction of the τm$^5$s$^2$U modification. Anticodon wobble base modifications in other mitochondrial tRNAs were also detected in our sequencing reads, but did not differ between wild-type and MERRF cells (Fig. 2b).

Next, we investigated whether there were any differences in the known m$^1$A modifications. To our surprise, the m$^1$A at position 58 of tRNA$^{Lys}$ (Fig. 1b) was absent in MERRF cells (Fig. 2a, c). It is important to note that mitochondrial tRNAs do not possess classical D and T arms [9] so the actual position of the m$^1$A in mitochondrial tRNA$^{Lys}$ is nucleotide 54 (m.8348 in the genome), however, we kept with conventional tRNA numbering for this modification. The m$^1$A58 is catalyzed by the methyltransferase 'writer' TRMT61B[23] and our quantitation of the modification by DM-tRNA-seq in wild-type cells is consistent with previous reports by primer extension and mass-spectrometry[23,24], which

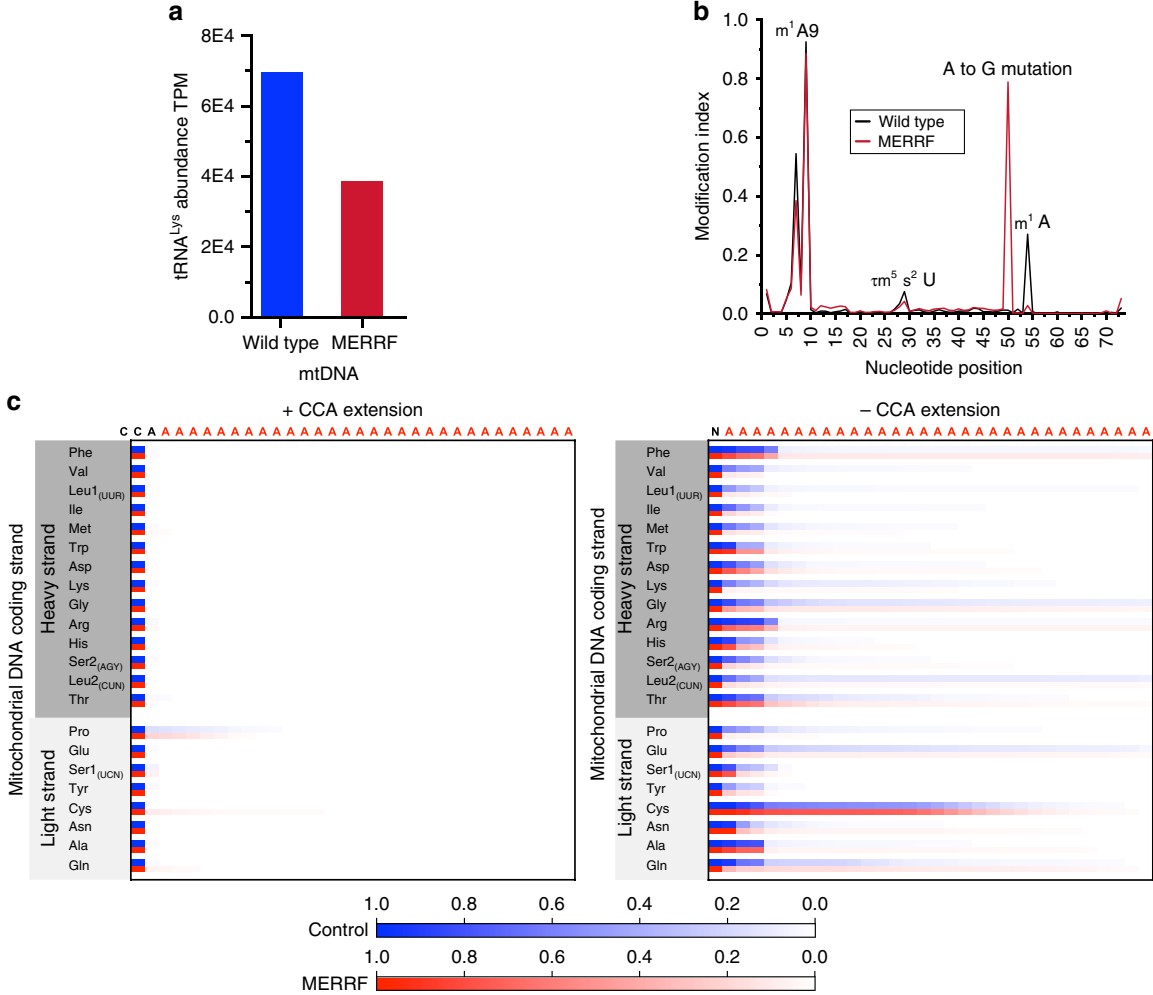

**Fig. 3** Novel RNA modifications of tRNA$^{Lys}$ linked to MERRF pathogenesis in human skeletal muscle. **a** Quantification of tRNA$^{Lys}$ abundance in skeletal muscle. **b** A position sequencing plot of tRNA$^{Lys}$ without demethylase treatment from skeletal muscle. **c** A 3′- position sequencing plot for the frequency of adenylation among mitochondrial tRNAs from skeletal muscle of a control and MERRF patient

demonstrate that this methyl modification was present only in a fraction of tRNA$^{Lys}$. In contrast, the m$^1$A9 on the D arm required for the formation of the cloverleaf structure of tRNA$^{Lys}$[25] and catalyzed by a different methyltransferase complex[26], was detected in both wild-type and MERRF cells. To test whether other known m$^1$A58 modifications in mitochondrial tRNAs were affected, we analyzed the m$^1$A58 on tRNAs Leu(UUR) and Ser (UCN), as well as m$^1$A16 in tRNA$^{Arg}$ and found no difference between wild-type and MERRF cells (Fig. 2c). Together, these data indicated that the loss of the m$^1$A modification in MERRF cells was restricted to tRNA$^{Lys}$ at position 58. Our sequencing approach expanded the landscape of MERRF pathogenesis by implicating a novel methyl modification on tRNA$^{Lys}$, highlighting the versatility and robustness of DM-tRNA-seq for investigating the regulation of mitochondrial gene expression disorders.

**m$^1$A58 tRNA$^{Lys}$ is linked to the MERRF pathogenesis in human skeletal muscle.** To assess the disease relevance of the modifications present in the tRNA$^{Lys}$ from proliferating myoblasts to the molecular pathogenesis of MERRF, we analyzed mitochondrial tRNAs from skeletal muscle biopsies obtained from a control and MERRF patient. Total RNA was isolated from control and MERRF muscle, and tRNA libraries were prepared for sequencing. From each of the libraries, 22–50 million reads

were mapped. The levels of nuclear-encoded tRNAs were strongly correlated between the two samples (Supplementary Figure 1B). By contrast, the steady-state level of tRNA$^{Lys}$ was ~ 50% lower in MERRF skeletal muscle than in the control (Fig. 3a), similar to our findings in myoblasts (Fig. 1e) (note that our cultured myoblasts originated from a different patient than the skeletal muscle biopsy). In skeletal muscle, the frequency of the MERRF mutation was approximately 80% (Fig. 3b), which is consistent with the threshold mutational load sufficient to cause a biochemical defect in oxidative phosphorylation[13].

To test the disease relevance of the post-transcriptional modifications on tRNA$^{Lys}$ to MERRF in skeletal muscle, we analyzed the modification index at positions 9, 34, and 58 (Fig. 1b) between the healthy control and patient. The cloverleaf stabilizing m$^1$A9 modification was found on the majority of tRNA$^{Lys}$ in both the MERRF patient and control (Fig. 3b). In contrast, the modification fraction of the m$^1$A58 in tRNA$^{Lys}$ was ~0.3 in control and absent in MERRF (Fig. 3b). This finding is entirely consistent with our data from proliferating myoblasts, strongly suggesting a key role of this modification in the pathogenesis of MERRF. The τm$^5$s$^2$U anticodon wobble base modification at U34 was modestly down in MERRF (Fig. 3b). Unfortunately, our analyses could not distinguish whether the m$^1$A58 co-exists *in cis* with the τm$^5$s$^2$U anticodon wobble base

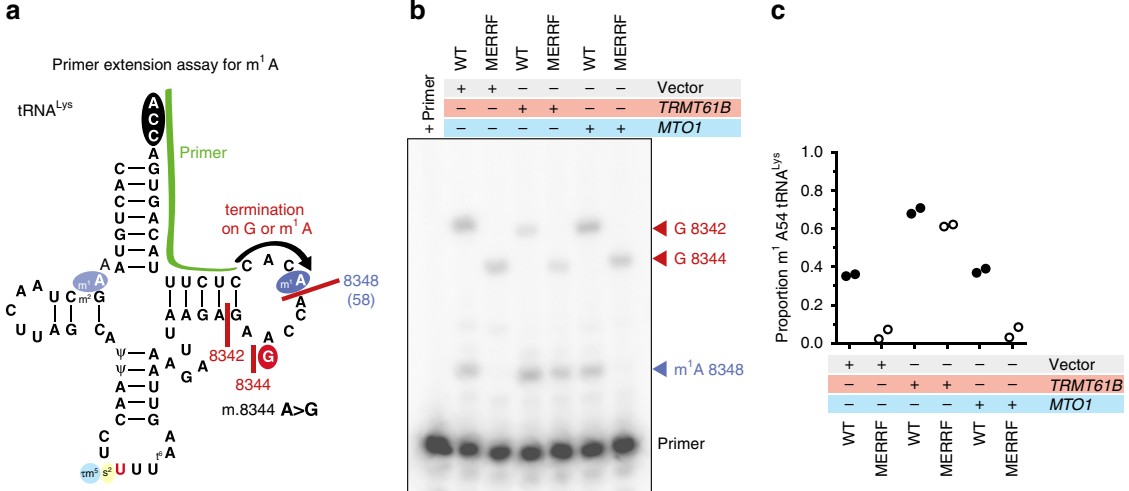

**Fig. 4** Restoration of the m$^1$A58 modification in tRNA$^{Lys}$. **a** Schematic of the primer extension assay for genotyping m$^1$A58 in tRNA$^{Lys}$. **b** A representative primer extension analysis on total RNA from human myoblasts stably transduced by retrovirus with the indicated cDNAs. **c** Quantification of primer extension analysis from two independent experiments. Proportion of m$^1$A58 was calculated as follows (m$^1$A 8348 / (m$^1$A 8348 + G 8342)) or (m$^1$A 8348 / (m$^1$A 8348 + G 8344))

modification. Together, our findings establish the relevance and expand the landscape of post-transcriptional RNA modifications on tRNA$^{Lys}$ to the molecular pathogenesis of MERRF.

A recent study demonstrated that aminoacylation of mitochondrial tRNAs bearing a CCA tail is regulated by oligoadenylation, catalyzed by the mitochondrial poly(A) polymerase (PAPD1); the steady-state level of this modification is governed by the balance between the activities of PAPD1 and phosphodiesterase 12 (PDE12), and is particularly prominent for tRNA$^{Lys}$ and tRNA$^{Ser2(AGY)}$[15]. Whereas another study showed how structurally abnormal mitochondrial tRNAs can be polyadenylated for degradation[27]. Therefore, we asked whether the adenylation mechanism is relevant to the pathogenesis of MERRF. For this analysis, we sorted sequence reads into two pools based upon the presence or absence of the CCA tail. Aminoacylation of tRNAs requires the addition of the CCA extension, which is catalyzed by a mitochondrially targeted TRNT1, and is critical for mitochondrial protein synthesis and human health[28,29]. When we analyzed the tRNA sequencing reads, oligoadenylation was widespread but only among the tRNA pool lacking the CCA extension in both control and MERRF (Fig. 3c). There was differential oligoadenylation between control and MERRF of selected mitochondrial tRNAs, including tRNA$^{Lys}$ (Fig. 3c). These data indicate that adenylation of mitochondrial tRNAs is a far more complex process than previously thought and may play a role in the regulation of mitochondrial tRNA populations and the pathogenesis of human disease.

**tRNA$^{Lys}$ RNA modifications have specific effects on translation.** Since the m$^1$A58 modification on tRNA$^{Lys}$ is consistently absent in MERRF myoblasts and skeletal muscle, we asked whether re-establishing the modification in the MERRF myoblasts could modulate mitochondrial translation. To this end, we retrovirally transduced myoblasts with the *TRMT61B* cDNA, reasoning that overexpression of this key methyltransferase would restore the modification on the tRNA$^{Lys}$. Indeed, *TRMT61B* overexpression restored the m$^1$A modification in MERRF tRNA$^{Lys}$ (Fig. 4), but did not alter the steady-state abundance of tRNA$^{Lys}$ (Supplementary Figure 2A). Importantly, TRMT61B overexpression increased the synthesis of selected proteins and

suppressed the generation of aberrantly sized polypeptides (Fig. 5a, b; Supplementary Figure 2B). The magnitude of this effect did not correlate with the number of lysine codons of a given polypeptide. (Supplementary Figure 2C).

Next, we asked whether the increased protein synthesis with *TRMT61B* overexpression also modulated the short term and long-term stability of mitochondrial nascent chains. Even though there was a 2-fold increase in $^{35}$S-incorporation in MT-CO1 and MT-ATP6 (Fig. 5b), these proteins were still unstable but with differential effects (Fig. 5c-e). By contrast, *TRMT61B* overexpression in the wild-type background led to a two-fold increase of the m$^1$A58 on tRNA$^{Lys}$ (Fig. 4c) and a selective dominant-negative effect specific to MT-CO1 nascent chain synthesis and long-term stability (Fig. 5a, b, e). Recent reports have also documented specific m$^1$A modifications in mitochondrial 12 S and 16 S rRNA, and mRNAs, including MT-CO1[30–32]. The m$^1$A modification at position 947 of the 16 S rRNA was proposed to be important for human mitochondrial ribosome subunit stability and is catalyzed by TRMT61B[30]. To test if mitochondrial ribosome stability was affected by TRMT61B overexpression, we assessed the steady-state levels of mitochondrial ribosomal proteins from the small and large subunit by immunoblotting because their abundance is a reliable proxy for stability[33] but found no difference (Fig. 5e). Our data showed that restoration of the m$^1$A58 on tRNA$^{Lys}$ can enhance translation in MERRF, arguing against this methyl modification as being a strictly negative regulator of mitochondrial gene expression as has been proposed for the m$^1$A modification on mitochondrial mRNAs[31,32]. Rather, there appears to be a tight regulation of this methyl modification on mitochondrial RNAs to finely tune mitochondrial protein synthesis.

Because reduced modification of the anticodon wobble base position is implicated in MERRF pathogenesis, we wanted to test whether overexpression of the key enzymes could also modulate mitochondrial translation. MTU1 catalyzes the thiol modification of the anticodon wobble base, a reaction that is required for human health[34], but appears to have no effect on mitochondrial protein synthesis in human fibroblasts[35]. Consistent with this finding, overexpression of *MTU1* in wild-type and MERRF myoblasts had no effect on mitochondrial protein synthesis (Supplementary Figure 2D and 2E). In contrast, overexpression of

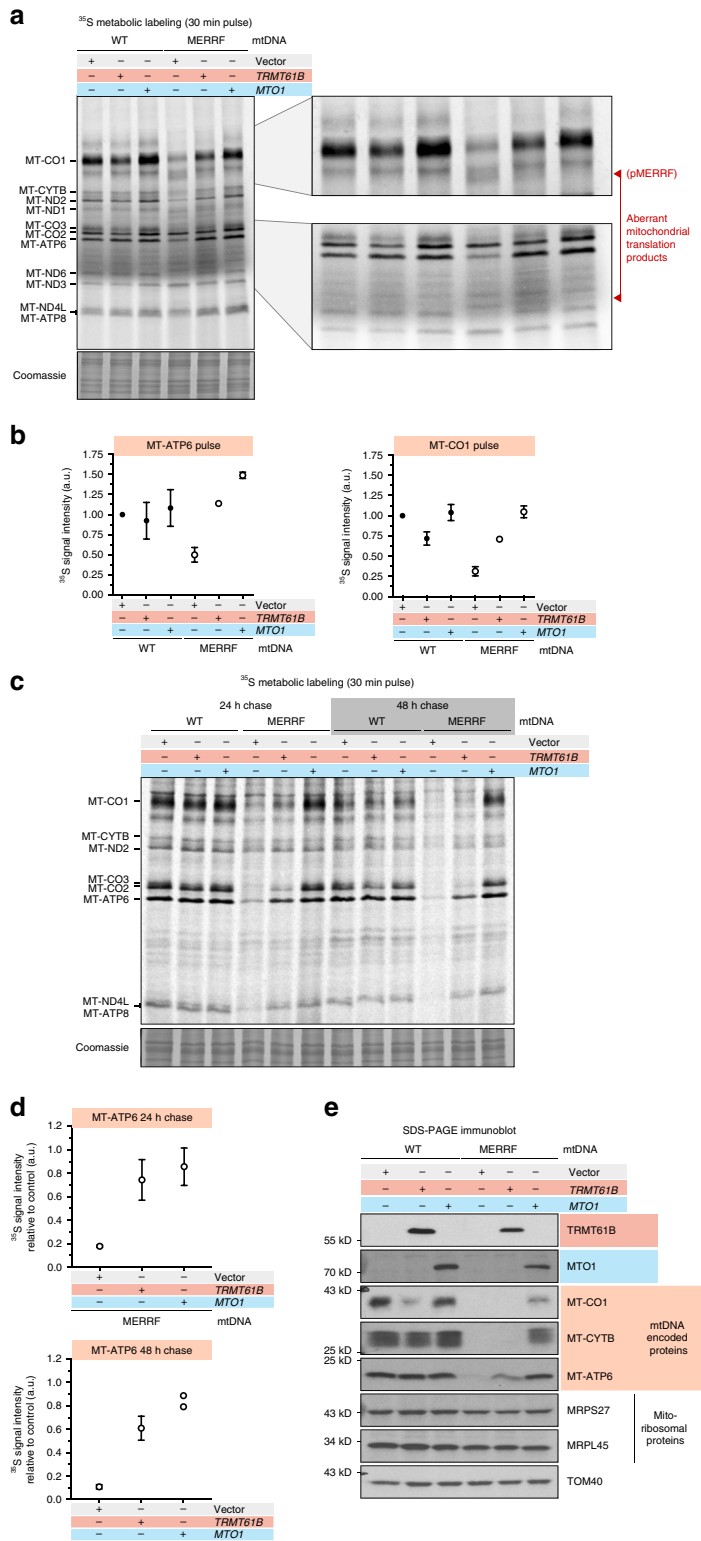

**Fig. 5** tRNA$^{Lys}$ RNA modifications reveal specific effects to translation fidelity. **a** A representative $^{35}$S pulse (30 min) metabolic labeling into mitochondrial protein synthesis of human myoblasts stably homoplasmic for the indicated mitochondrial DNA transduced by retrovirus with the indicated cDNAs. Aberrantly sized labeled polypeptides are indicated. **b** Quantification of $^{35}$S incorporation into selected mitochondrial proteins during a 30 min pulse (from **a**). Data are represented as mean +/− S.D. from three biological experiments. **c** A representative 24 and 48 h cold chase following a 30 min $^{35}$S pulse metabolic labeling of mitochondrial protein synthesis in human myoblasts. **d** Quantification of MT-ATP6 stability in the chase relative to wild-type cells transduced with an empty vector. Data is mean +/− S.D. from three biological experiments except for MTO1 at 48 h, where only the data from two independent experiments are shown. **e** Immunoblotting of whole-cell lysates from human myoblasts homoplasmic for the indicated mitochondrial DNA transduced by retrovirus with the indicated cDNAs decorated with the indicated antibodies. Representative data of multiple independent experiments

*MTO1* had a marked effect on mitochondrial protein synthesis and protein stability (Fig. 5). MTO1 forms a complex with the small GTPase GTPBP3 to catalyze the 5-methylaminomethyl (mnm$^5$U34) modification on the anticodon wobble base found in mitochondrial tRNAs of Glu, Lys, Gln, Leu$^{(UUR)}$, and Trp[9]. *MTO1* overexpression restored mitochondrial protein synthesis to wild-type levels (Fig. 5a, b), but there was a differential effect on the stability of individual proteins (Fig. 5c–e) that did not correlate with the abundance of lysine codons in a given polypeptide (Supplementary Figure 2C). Moreover, these effects were independent of the m$^1$A58 modification (Fig. 4b, c). Although the rate of MT-CO1 synthesis increased, these nascent chains remained unstable in contrast to MT-ATP6. No adverse effects were observed on wild-type myoblasts overexpressing *MTO1*. Collectively, these findings demonstrate that post-transcriptional modifications of mitochondrial tRNAs exert differential effects on protein synthesis and stability, revealing the profound influence that RNA modifications have on mitochondrial gene expression and the pathogenesis of human disease.

## Discussion

RNA modifications have emerged as major regulators of gene expression across all domains of life[36]. In this study, we showed that post-transcriptional modifications on a single mitochondrial tRNA modulate nascent chain synthesis and protein stability. Our data also highlight the importance of using diploid patient cells to investigate the regulation of mitochondrial gene expression in the pathogenesis of mitochondrial diseases.

Our data show that the MERRF tRNA$^{Lys}$ mutation simultaneously impacts modifications at the wobble anticodon position (τm$^5$s$^2$U) and m$^1$A58 in the T-loop. Although the effect of the MERRF mutation on τm$^5$s$^2$U was previously established[5], the complete elimination of m$^1$A58 in the MERRF mutation was unexpected. m$^1$A58 introduces a positive charge in the elbow region of the tRNA tertiary structure, which can affect the binding of the elongation factor that delivers tRNA to the ribosome[37]. Therefore, weakened binding of the m.8344 A > G mutant tRNA$^{Lys}$ to mitochondrial EF-Tu due to elimination of m$^1$A58 may also contribute to the disease phenotype. Consistent with this interpretation, overexpression of the m$^1$A58 methyltransferase, TRMT61B, restored the modification, increasing the synthesis of mitochondrial nascent chains and preventing the generation of aberrantly sized polypeptides. In fact, restoring the methyl modification on tRNA$^{Lys}$ completely rescued the translation defect in MT-ATP6 synthesis. Together, these data argue that the m$^1$A58 modification is a positive regulator for mitochondrial translation elongation. Despite rescuing the translation defect, however, most nascent chains were still unstable. Strikingly, the stability of MT-ATP6 and MT-ATP8 was completely different from the other mitochondrial nascent chains, a phenotype that did not correlate with the number of lysine codons within in the mRNA. These data point to a more complex molecular pathogenesis of the m.8344 A > G tRNA$^{Lys}$ mutation.

It is curious that the m$^1$A58 is found only in a subset of tRNA$^{Lys}$ molecules whereas for the tRNA$^{Leu(UUR)}$ >90% of the tRNAs possess this methyl modification. In a canonical tRNA fold such as tRNA$^{Leu(UUR)}$ the m$^1$A58 forms a reverse-Hoogsteen base pair with 5-methyluridine or adenosine at position 54 in the T-loop, a m$^1$A58-A54 interaction may also form in tRNA$^{Lys}$ and the m$^1$A modification may affect the stability of this interaction too. In contrast, the frequency of the cloverleaf inducing m$^1$A9 modification is similar between tRNA$^{Lys}$ and tRNA$^{Leu(UUR)}$. Absence of the post-transcriptional modification of the wobble anticodon position of the cytoplasmic tRNA$^{Lys}$ has been shown to

cause ribosome pausing on lysine codons, inducing a severe delay in the rate of translation elongation[10–12]. The data from other translation systems suggest a uniform rate of decoding for all tRNAs[38]. The main differences appear to be in the stability of the codon recognition complex among tRNAs, which specifically affect the geometry of the codon–anticodon complex and dis-association rates[12]. Consistent with this interpretation, over-expressing *MTO1* dramatically increased the rate of mitochondrial protein synthesis, but most of the nascent chains were still unstable, suggesting the fidelity of mitochondrial protein synthesis is still disrupted. Our data suggest the m.8344 A > G MERRF mutation causes protein misfolding at two levels: one, from decreased rates of translation elongation and two, from a decreased fidelity of protein synthesis. Together, our data point to two post-transcriptional modifications on the tRNA$^{Lys}$ that are important for the efficiency and fidelity of mitochondrial protein synthesis. Since the 13 mitochondrial proteins are co-translationally inserted into the lipid bilayer to assemble into large heterooligomeric complexes[39], nascent chain misfolding would interfere with the kinetics of assembly for the oxidative phosphorylation complexes.

A recent study found that the tRNA$^{Lys}$ m$^1$A58 methyltransferase TRMT61B is also responsible for m$^1$A modifications in several mitochondrial mRNAs[31]. Upon *TRMT61B* over-expression, the m$^1$A level in mitochondrial *MT-CO2* and *MT-CO3* mRNAs increased, but the steady-state levels of these proteins decreased, leading to the conclusion that m$^1$A in mitochondrial mRNAs is a negative regulator of protein synthesis[31]. The methyltransferase TRMT10C, which is responsible for the m$^1$A9 modification in tRNAs, can also post-transcriptionally modify mitochondrial mRNAs, in particular *MT-ND5*, in a process that is developmentally regulated during embryogenesis and post-natally between germline and somatic tissues[32]. Since m$^1$A on mitochondrial mRNAs would interfere with Watson–Crick base pairing, this modification would be inhibitory in mRNA during translation. In contrast to the mRNA m$^1$A effects, our results show that m$^1$A58 in tRNA$^{Lys}$ strongly increases protein synthesis, and this tRNA modification is also important for the stability of MT-ATP6. These results suggest that m$^1$A modifications catalyzed by the same methyltransferase could have opposing effects on mitochondrial protein synthesis: an inhibitory role on mRNAs and positive role with mitochondrial tRNA. Such a mechanism on mitochondrial mRNAs might be a part of a quality control step to ensure aberrant mRNAs are not fully engaged in translation to minimize the potential for misfolded proteins. These opposing effects may explain the sub-stoichiometric m$^1$A58 level in tRNA$^{Lys}$ in wild-type cells. An elevated m$^1$A58 level on tRNA$^{Lys}$ may compete with decoding of other tRNAs at near-cognate codons, thereby decreasing translational fidelity and ultimately promoting protein misfolding and degradation. The optimal modification level could be maintained by the activities of the writer, TRMT61B, and the mitochondrial tRNA m$^1$A eraser ALKBH1[24]. The expression level of these enzymes could vary across different cell types and tissues, and be a key regulator for modulating mitochondrial protein synthesis. In the cytosol, dynamic regulation of the m$^1$A modification on tRNAs in response to metabolic cues is a significant regulator of cellular protein synthesis[37]. Therefore, it is intriguing to speculate that a similar metabolic regulation of this post-transcriptional modification exists for mitochondrial gene expression.

An interesting finding of our study points to different pools of tRNAs within the organelle. This effect is exemplified by the oligoadenylation of molecules lacking the CCA extension, which implies the existence of separate pools of tRNAs with and without this post-transcriptional polyA extension. A previous report found that deficiency of PDE12, the phosphodiesterase that

removes oligoadenylates from mitochondrial RNA, triggers oligoadenylation of tRNA$^{Lys}$ containing a CCA extension, which was proposed to be a regulator step for aminoacylation[15]. The physiological significance of these distinct tRNA populations remains unclear. Polyadenylation status in mitochondrial mRNA is largely positively correlated with stability and translational competence[40], and defects in the enzyme responsible for this modification, PAPD1, are linked to spastic ataxia in humans[41]. In *Drosophila*, the mitochondrial poly(A) polymerase (MTPAP) is required to protect the integrity of mRNA 3′ ends, but not for stability or translation[42]. Loss of this enzymatic activity in flies inhibits the maturation of tRNA$^{Cys}$, a very rare codon in mitochondrial mRNAs (1.1% in flies; 0.6% in humans). Moreover, a recent report suggests structurally unstable tRNAs are polyadenylated as part of a degradation step[27]. Our ability to distinguish different pools of mitochondrial tRNA was only possible because of the ability of DM-tRNA-seq to obtain nucleotide resolution. Future studies should seek to systematically address the dynamic regulation of 3′ adenylation and its consequences on tRNA stability and maturation.

Equally surprising was the discovery that RNA modifications have distinct effects on the stability and synthesis of the 13 mitochondrial proteins. These observations highlight the limitations of our current understanding of mitochondrial protein synthesis and its regulation. The application of new RNA sequencing approaches to this organelle should provide insight into these mechanisms, as well as their contributions to human disease.

## Methods

**Cultured cells.** Homoplasmic human myoblasts for wild-type or the m.8344 A > G tRNA$^{Lys}$ mutation were derived from satellite cells from a skeletal muscle biopsy[13] and immortalized with E6/E7[43] and hTERT. Cells were cultured at 37 °C and 5% CO$_2$ in Cell Applications Inc. Skeletal Muscle Cell Growth Medium (#151–500) supplemented with 50 μg/ml uridine. Retrovirus was generated by transient transfection of retroviral plasmids into the Phoenix amphotropic packaging line as in[44]. Cells were used directly following selection with blasticidin. All cells were tested for mycoplasma before use.

**Human skeletal muscle biopsy.** The human control skeletal muscle biopsy was obtained after informed consent, according to Helsinki Declaration, accepted by the Ethical Review Board of Helsinki University Hospital (nr 43/13/03/04/2008). The MERRF skeletal muscle biopsy was obtained after informed consent from a subject with progressive muscle weakness. The skeletal muscle biopsy showed accumulation of COX negative and ragged-red muscle fibers and a high level of heteroplasmy for the mutation m. 8344 A > G in tRNA$^{Lys}$.

**Cloning.** Wild-type cDNAs (TRMT61B, clone BC010365.1; MTU1 (TRMU), clone CU013043; MTO1, clone BC011051) were obtained from the ORFeome or Mammalian Genome Collection. These cDNAs were in a Gateway entry vector (pENTR221 or pCMV-SPORT6) or cloned into pDONR201 using PCR with KAPA HiFi followed by recombination. The *TRMT61B* cDNA was missing a stop codon, which was added by PCR using the primers BB-477 (5′-aaaaagcaggctac-cATGCTAATGGCATGGTGCCG-3′) and BB-478 (5′-agaaagctgggtttattaGT-TAAGTTGTGGTTTGACC-3′) with KAPA HiFi followed amplification with attB1 and attB2 adapters then by gateway cloning into pDONR201. All full-length cDNAs were recombined into Gateway converted pMXs-IRES-Blasticidin with LR Clonase II (Thermo Fisher). All cDNAs were thoroughly sequenced by Sanger sequencing for verification initially and after all PCR manipulations.

**Rna isolation.** Total RNA was isolated from cultured myobalsts (homoplasmic for wild-type mtDNA or the MERRF mtDNA mutation) from three independent cultures with Trizol according to the manufacturer's instructions. Muscle samples were homogenized with a Precellys 24 (Bertin Technologies). For next-generation sequencing, all samples were reprecipitated with 0.1 volume 3 M Sodium acetate and 3 volume ice cold 100% Ethanol.

**DM-tRNA-seq.** DM-tRNA-seq was performed as previously described[17]. Briefly, purified RNA was mixed with three T7 transcribed tRNA standards (0.01pmol each standard per μg total RNA). Deacylation of total RNA (at < 0.5 μg/μL) was performed in 100 mM Tris-HCl pH = 9.0 at 37 °C for 30 min. Small RNA was

isolated from total RNA by using the small RNA isolation protocol for the RNA Clean and Concentrator Kit (Zymo R1015).

Up to 1.5 μg of small RNA (~60 pmol) was demethylated with 2× molar ratio WT AlkB and 4X molar ratio of D135S AlkB in 25 mM MES pH = 5.0, 30 mM KCl, 2 mM MgCl$_2$, 2 mM ascorbic acid, 300 μM α-ketoglutarate, 50 μM (NH$_4$)$_2$Fe (SO$_4$)$_2$ in the presence of RNasin at room temperature for 2 h. Demethylated small RNA was purified using an RNA Clean and Concentrator Kit (Zymo). Next, 3′-phosphate removal was performed on both demethylase-treated and -untreated small RNA. Small RNA was treated with T4 PNK (Affymetrix, final concentration 0.1U/μL) in 1× T4 PNK buffer at 37 °C for 30 min. The RNA Clean and Concentrator (Zymo) was again used to clean up the reaction.

For cDNA synthesis, the TGIRT DNA primer was 5′ labeled with T4 PNK. 4pmol of 5′ labeled TGIRT primer (5′-GATCGTCGGACTGTAGAACTAGACGTGTGCTCTTCCGATCTT-3′) was annealed to 4pmol of complementary RNA (5′-AGAUCGGAAGAGCACACGUCUAGUUCUACAGUCCGACGAUC/3SpC3/-3′) in 100 mM Tris-HCl pH = 7.5, 0.5 mM EDTA at 82 °C for 2 min, then slow cooled to room temp. Up to 100 ng small RNA (~4 pmol tRNA) was then added. The small RNA/primer mixture (~200 nM) was then preincubated at room temp for 30 min in 100 mM Tris-HCl pH = 7.5, 450 mM NaCl, 5 mM MgCl$_2$, 5 mM DTT with 500 nM TGIRT (InGex, Inc.). dNTPs were added to 1 mM, and the reverse transcription reaction was carried out at 60 °C for 60 min. NaOH was added to 0.25 M and the mixture was heated at 95 °C for 5 min to hydrolyze RNA. HCl was added to 0.25 M to neutralize the reactions.

An equal volume of 50% formamide, 4.5 M urea, 50 mM EDTA, 0.05% Bromophenol blue, 0.05% xylene cyanol was added to the cDNA, and the mixture was heated at 95 °C for 15 min. cDNA was then gel purified by denaturing 10% PAGE (7 M urea, 1X TBE), and extended products were excised from the gel and eluted overnight at room temperature in 200 mM KCl, 50 mM KOAc. Purified cDNA was ethanol precipitated with addition of linear acrylamide (Thermo) to 20 μg/mL.

Purified cDNA was circularized using CircLigase II (Epicentre) at 60 °C overnight. The enzyme was inactivated at 80 °C for 10 min, and samples were then phenol/chloroform extracted and ethanol precipitated. Illumina libraries were prepared using Phusion Master Mix (Thermo) for 12 PCR cycles (98 °C 5 s, 60 °C 10 s, 72 °C 10 s). AMPure XP Beads (Beckman-Coulter) were used to clean up the libraries before Illumina sequencing. All libraries were sequenced on an Illumina HiSeq 2000 with paired-end mapping using read lengths of 100 base pairs.

**tRNA-seq data analysis.** Alignment and analysis of tRNAs at single-base resolution was performed similarly as in ref. [18]. Briefly, paired-reads were processed using Trimmomatic v 0.32n and custom Python scripts to remove traces of adapter. For the purpose of poly-A tail analysis, low quality homo-nucleotide stretches were preserved. Processed reads were aligned in Bowtie 1.0 according to the parameters: –v 3 –m 10 –best –strata. Fragments shorter than 15 base pairs during alignment were removed in subsequent analysis, and further C and Python scripts merged reads onto individual tRNAs. For poly-A tail analysis, additional remapping was performed by isolating reads and aligning them onto reference genomes with poly-A(30) appended. Sequence mapping of human mitochondrial-encoded tRNAs is straightforward because each mt-tRNA sequence is unique and divergent from the nuclear-encoded tRNAs.

Modifications are measured as the sum of mutation fractions and the RT stop fraction for any given tRNA seed sequence at each nucleotide position. This sum of mutation and stop fraction is presented as a Modification Index (MI) value which is between 0–1 for each position. The DM-tRNA-seq method does not have a fragmentation step prior to cDNA synthesis, and all sequencing reads start from the 3′ CCA[17]. Therefore, stop fraction can be accurately mapped for the tRNA positions more than ~25 nucleotides away from the 3′ end. For the tRNA positions close to the 3′ end, stop fraction mapping is much less accurate because the cDNA products generated by RT stop can be too short to be in cluded in the sequencing library construction. For these positions, which can include A58 of mitochondrial tRNA$^{Lys}$, the MI value would represent a lower bound of the modification fraction.

Our MI-value based quantitation of m$^1$A modification compares well with those determined by mass spectrometry. For HEK293T cells, mass-spectrometry results from Suzuki and co-workers[24] shows a modification fraction of 17.9% for m$^1$A16 in mitochondrial tRNA$^{Arg}$ and 48.3% for m$^1$A58 in mitochondrial tRNA$^{Lys}$. The MI value determined by DM-tRNA-seq[18] is 0.16 for m$^1$A16 of mt-tRNA$^{Arg}$ and m$^1$A58 of 0.24 for mt-tRNA$^{Lys}$. m$^1$A16 is 52 nucleotides away from the 3′ end, its MI includes both mutation (0.074) and stop (0.086). m$^1$A58 is 20 nucleotides away from the 3′ end, its MI shows only the mutation fraction, therefore, the modification fraction is likely higher. Both m$^1$A16 and m$^1$A58 have the same sequence context of m$^1$ACA so that a similar level of mutation and RT stop ratios may be present in regards to context dependence[45]. Therefore, the m$^1$A58 modification fraction can be estimated to be around 0.48 which is similar to the mass spectrometry result.

For statistical analysis of tRNA-seq, the data were analyzed were indicated with paired sample $t$ test.

**Primer extension assay.** RNA oligonucleotide probes (tRNA$^{Lys}$ 5′-TGGTCACTGTAAAGAG-3′, where TGG correlates to CCA) were 5′ labeled with

gamma $^{32}$P ATP by T4 Polynucleotide Kinase (NEB) according to the manufacturer's instructions. Labeled oligonucleotides were separated from non-incorporated nucleotides via G40 MicroSpin columns (GE Healthcare). A concentration of 4 μg of total RNA from patient and control myoblasts was annealed to 0.36 pmol of radiolabeled specific primer in 30 mM Tris-Cl pH 7.5 and 2 mM KCl at 90 °C for 2 min and then cooled down to room temperature for 5 min. Reverse transcription reaction with 0.2 U/μL AMV Reverse Transcriptase (NEB) was performed at 42 °C for 30 min using a dNTP/ddCTP mix at 0.25 mM each (final concentration). Subsequently, RNaseH (Epicentre) was added, and the reaction incubated at 37 °C for 10 min to degrade RNA. To 5 μL cDNA, 5 μL RNA Loading dye (Formamide + bromophenol blue) was added and RT products resolved on a 12% denaturing PAGE (7 M Urea,1× TBE, 30 cm). After the run, the gel was fixed and dried on whatman paper and exposed to a Phosphor screen.

**Semi-quantitative RT-PCR**. Total RNA was isolated from myoblasts with the Monarch total RNA Miniprep Kit (NEB) with on-column DNAseI digest. A gene-specific primer (5′-GGCACTCCATCATCTTGTCC-3′ for *MTU1* and 5′-CTTCCAGGCTGGGACTGAG-3′ for *ATAD3A*) was annealed to 400 ng of RNA and reverse transcribed with Maxima H Minus RT (Thermo Scientific) at 50 °C for 30 min. For the subsequent PCR, 5% of the RT-reaction was used as a template with Econo-Taq (Lucigen) and primers specific for *MTU1* (forward 5′-CAGGTTTCCCAGGATGCCC-3′, reverse 5′-AGGAACCAACCTT-TATGTGTTCC-3′) and *ATAD3A* (forward 5′-TGAGGAGAGGAGGAAGACCC-3′, reverse 5′-TGCTGCTTCAGTTGGTCCTC-3′).

**Immunoblotting**. Cells were prepared as described in ref. [44]. Briefly, whole-cell pellets were solubilized in phosphate buffered saline, 1% dodecyl-maltoside (DDM), 1 mM PMSF (phenylmethylsulfonyl fluoride), and complete protease inhibitor (Thermo Fisher). Protein concentrations were measured by the Bradford assay (BioRad). Equal amounts of proteins were separated by Tris-Glycine SDS-PAGE. Membranes were blocked in TBST with 1% milk at room temperature for 1 h then primary antibodies (Proteintech Group: MTO1 (15650-1AP, 1:5000); TRMT61B (26009-1AP, 1:2000); MT-CYB (55090-1-AP, 1:1000); MT-ATP6 (55313-1-AP, 1:2000); MRPL11 (15543-1-AP, 1:20000); MRPL45 (15682-1-AP, 1:5000); MRPS27 (17280-1-AP, 1:5000). Abcam: MT-CO1 (1D6E1A8, 1:1000); SDHA (2E3GC12FB2AE2, 1:10000). Santa Cruz: TOM40 (sc-11414, 1:2000)) were incubated overnight at + 4 °C in 5% BSA/TBST and detected the following day with secondary HRP conjugates (Jackson ImmunoResearch) using ECL with film. Representative data of independent experiments were cropped in Photoshop with only linear corrections applied.

**Metabolic labeling of mitochondrial protein synthesis**. Mitochondrial protein synthesis was analyzed by metabolic labeling with $^{35}$S methionine/cysteine[44]. Cells were pre-treated with anisomycin (100 μg/ml) to inhibit cytoplasmic translation then pulsed with 200 μCi/ml $^{35}$S Met-Cys (EasyTag-Perkin Elmer). In chase experiments, cells were pulsed for 30 min with radiolabel then the medium removed and replaced with fresh medium lacking the radioisotope for the indicated time. Equal amounts of sample protein were first treated with Benzonase (Thermofisher) on ice and then resuspended in 1x translation loading buffer (186 mM Tris-Cl pH 6.7, 15% glycerol, 2% SDS, 0.5 mg/ml bromophenol blue, 6% beta-mercaptoethanol). A 12–20% gradient Tris-Glycine SDS-PAGE was used to separate samples then dried for exposure with a Phospho screen and scanned with a Typhoon 9400 or Typhoon FLA 7000 (GE Healthcare) for quantification. Gels were rehydrated in water and Coomassie stained to confirm loading.

## Data availability
 All tRNA-seq data are deposited to NCBI GEO with the accession number GSE106601 and all other data is available from the authors.

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

## Acknowledgements

We thank L.S. Churchman, H.T. Jacobs, and M. Toompuu for valuable discussion; K.Y. Ng for technical help; and C. Patil for editing. This work was supported by the Academy of Finland Centre of Excellence on Mitochondria to B.J.B.; an Academy of Finland postdoctoral award to U.R.; by the National Institutes of Health [RM1HG008935 to T. P.]; M.E.E. and W.C.C. were supported by the National Institutes of Health Chemistry and Biology Training Grant [T32 GM008720]; and M.E.E. was a recipient of the National Science Foundation pre-doctoral fellowship [DGE-1144082].

## Author contribution

U.R., M.E.E., T.P. and B.J.B. conceived and designed the research. U.R., M.E.E., W.C.C., and P.M. performed all of the experiments. U.R., M.E.E., W.C.C., P.M., T.P. and B.J.B. analyzed the data. A.S., A.We., A.Wr. provided critical reagents. E.A.S. provided critical reagents and editing of the manuscript. B.J.B. and T.P. wrote the manuscript and all authors commented on it.

## Additional information

**Competing interests:** The authors declare no competing interests.

