## [Peer Review File · Nature Communications]

Reviewers' comments:

Reviewer #1 (Remarks to the Author):

The authors argue for a previously overlooked m1A modification at the MERRF mutational position in tRNA Lys (position 8344) in patient cells. They show an impact on tRNA stability, and resultant mitochondrial protein synthesis. Although previous reports demonstrated the impact of the MERRF mutation on m5s2U modification, the complete elimination of m1A54 in the MERRF mutation was not reported. In general, I was impressed by the amount of analyses presented and the potential interest for researchers in the fields of mitochondrial biology, physicians, and in general – people that are interested in RNA modifications. Nevertheless, there are some major concerns that need to be addressed prior to further consideration of the manuscript for publication.

Major comments:

1. Firstly, Chujo and Suzuki reported in 2012 (Chujo and Suzuki (2012) *RNA*; 18(12): 2269–2276) a partial m1A modification of tRNA^{Lys} position 58, which interacts with position 54 at the T-loop. They did not observe any Wild Type modification at position 54 of mitochondrial tRNA^{Lys}, but rather at position 58 (Figure 3C in Chujo and Suzuki, 2012). Is it possible that the authors of the current manuscript observed the previously described m1A at position 58 of tRNA^{Lys}? Position 54, which is mutated in MERRF (human mtDNA position 8344), and was claimed to undergo an m1A modification in wild type muscle cells, was never shown to undergo an m1A modification by any other group who studied the m1A modification of this tRNA. I find this very strange, especially, since the authors chose to totally ignore this discrepancy throughout the manuscript even while citing the 2012 paper by Chujo and Suzuki. This issue has to be carefully attended by the authors prior to further consideration of the manuscript.
2. In lines 143-144 the authors wrote: "DM-tRNA-seq allowed us to assess quantitatively differences in specific RNA modifications across all mitochondrial tRNAs between wild-type and MERRF-mutant cells". This method (Zheng et al *Nature Methods* 2015) is only relatively quantitative, while absolute quantification seems challenging. In the current manuscript the authors calculated modification index, which is OK, although again reflects relative abundance of the modifications, rather than absolute numbers. This should be clarified to avoid over interpretation of the data.
3. In line 176 the authors wrote a section entitled "m1A54 tRNA^{Lys} is linked to the MERRF pathogenesis in human skeletal muscle". This relates to major comment #1: Since previously an m1A modification was reported in tRNA^{Lys} position 58, and since this m1A at position 58 forms a reverse-Hoogsteen base-pairing with 5-methyluridine (m5U) at position 54 in the T-loop of tRNAs, it is very conceivable that the MERRF mutation affects such interaction, and in turn, affects the modification's level. I feel that this is what the authors observed in MERRF cells. Additionally, since the MERRF mutation is a point mutation, it is not surprising that other mtDNA recognition sites of TRMT61B are not affected, thus explaining why the levels of other m1A modifications throughout the mtDNA are unchanged. This interpretation should be recognized by the authors and the text throughout the manuscript should be altered accordingly.
4. In lines 249-251 the authors wrote: "MTU1 catalyzes the thiol modification of the anticodon wobble base, a reaction that is required for human health (23), but appears to have no effect on mitochondrial protein synthesis in human fibroblasts (24)". The authors nicely cite differential effect of MTU1 driven thiol modification, but do not refer to tissue differences in mitochondrial m1A modifications, which are shown in Safra et al 2017 (they cited this paper elsewhere in the manuscript). As far as I am concerned there could be tissue differences in the level of the m1A modifications, which should be discussed in the current manuscript while analyzing only muscle cells.

Minor comments:

1. In lines 120-122 the authors wrote: "Within the mitochondrial genome, the genes encoding these two tRNAs flank the 12S rRNA gene, and are transcribed together, along with the 16S rRNA,

as a short polycistronic RNA at a level 25-fold higher than those of genes downstream along the H-strand (14)". The authors cited the 1993 paper by King and Attardi (ref 14 - J. Biol. Chem. 268, 10228–10237). A more recent report (Blumberg et al (2017) Genome Res;27(3):362-373) showed by using a quantitative in vivo nascent RNA transcription detection assay that there were tremendous quantitative differences in the transcription ratio between the light and heavy strands in various human cell lines, including HeLa cells – very different numbers than reported in the King and Attardi paper. The authors are asked to take into account the reported levels of transcription in this latter report (Blumberg et al. 2017) and change their statements accordingly.

2. In lines 238-242 the authors wrote: "The m1A modification at position 947 of the 16S rRNA was proposed to be important for human mitochondrial ribosome subunit stability (20). Our analysis of the stability of mitochondrial ribosomal proteins from the small and large subunit indicated no difference following TRMT61B overexpression (Figure 4F)". I went and read citation 20 (Bar-Yaacov et al. PLoS Biology, 2016), and this paper suggested that the modification is presumably important to stabilize the structure of the mito-ribosome, not the levels of the mitochondrial ribosomal proteins. Thus, although a nice finding, I do not see any discrepancy between the findings of the current manuscript and citation 20. This should be acknowledged.

Reviewer #2 (Remarks to the Author):

The authors present data analyzing the biology of MERRF, a well described mitochondrial disease due to a mutation in the T-loop of mitochondrial tRNA^{Lys}(UUU), which leads to reduction in 5-taurinomethyl-2-thiouridine. They find evidence that human diploid myoblast cells homoplasmic for the mutation have aberrant translation, inconsistent with the expectation of reduced translation due to ribosome stalling from decoding problems. They then use their tRNA sequencing methods (DM-tRNA-seq) and show that tRNA^{Lys} is reduced in these MERRF diploid cells; provide unexpected evidence that tRNA abundance does not correlate with codon usage, and provide evidence that the nuclear genome differentially regulates the mitochondrial genome. By using a newly developed methodology for quantifying modifications, the authors provide evidence that MERRF cells have reduced 5-taurinomethyl-2-thiouridine for tRNA^{Lys} (but not other tRNAs with this modification), and, unexpectedly, reduced m1A58 in tRNA^{Lys} (but not in other tRNAs with this modification). Analysis of skeletal muscle cells from another MERRF patient corroborated the conclusion that tRNA^{Lys} was reduced and that m1A58 was absent in the patient cells, while 5-taurinomethyl-2-thiouridine was only reduced modestly. Oligoadenylation was observed in tRNA^{Lys} from MERRF cells but not from WT cells or from tRNA^{LeuUUR}, but the reduction was only at internal positions and not from the CCA end, arguing against a charging defect regulated by oligoadenylation. Overexpression of TRMT61B (encoding a subunit of the m1A MTase) results in WT levels of m1A58, but not increased levels of tRNA^{Lys}, and improved mitochondrial translation but not improved stability of the proteins. Similar results were observed by overproducing MTO1, encoding the enzyme involved in formation of the taurinomethyluridine moiety, again resulting in more translation, but not more stability.

Although the topic of this manuscript is important and appropriate for this journal, the material at this time is not as well developed as one would like, and the bottom line seems obscure. Moreover, there are significant issues with aspects of the data and interpretation that are incomplete as presented. There is no doubt in this reviewer's mind that m1A formation is reduced in MERRF cells, as claimed, but some of the other conclusions, are more difficult.

1. The title of the manuscript does not accurately reflect its content; it is not about the modification landscape, but is entirely about MERRF. Similarly, the abstract inadvertently implies that m1A58 of tRNA^{Lys} is important for stability of proteins, by the phrasing "By restoring the modification on the mitochondrial tRNA^{Lys}, we demonstrated the importance of the m1A54 to mitochondrial protein synthesis and nascent chain stability."

2. The data in Fig. 1B is mischaracterized, in the opinion of this reviewer. It seems clear that

mitochondrial translation is almost completely defective in the MERRF cells, and the aberrant translation products are a minor but interesting side issue. The argument in the text lines 68-74 seems unfounded by the data presented, since translation looks defective, as expected. Moreover, there is no set of controls to validate that the aberrant translation in Fig. 1B is from the mitochondrial proteins. In this figure the red and black stars are not explained.

2. The data in Fig. 1E and 1F seems ancillary to this reviewer, and if included, should contain a comparison to other known analyses for cytoplasmic tRNAs in other organisms to make a point. Moreover, in this reviewer's opinion the comparison to genomic codon usage is not nearly as relevant as that to highly expressed proteins, to mRNA levels, or some sort of codon adaptation index. In addition a trend line and R value or R-squared value would add credence to the claim of lack of correlation.

4. The authors should define the wobble mutation in tRNA^{Lys} by name (5-taurinomethyl-2-thiouridine), and should explain what heavy strand and light strand mean to readers not well versed in mitochondrial biology.

5. The data in Fig. 4B,E, and F is somewhat confusing. Fig. 4B and 4F appear to be from the same experiment or set of experiments, based on the legend, but 4F and 4E are from the same experiment based on the text.

6. It would be very useful to see a Western blot for the data in Fig. 4B to more accurately quantify or visualize translation provoked by overexpression of TRMT61B and MTO1 (instead of, or in addition to, Fig. 4C)

7. It would be important to have a more complete time course of the instability of the proteins in Fig. 4E,F, accompanied by a comparison to the doubling time of the cells. This could help determine if and to what extent increased m1A (or 5-taurinomethylthiouridine) on the tRNA (and mRNA) impacts protein half-life relative to controls.

8. I am not sure why this manuscript shows dynamic regulation of modifications (line 305). It is true that modifications are dynamic, but the evidence here that they are is not that obvious.

9. The discussion may need some focusing on the possibility that translation rates impact folding and stability of proteins.

Reviewer #3 (Remarks to the Author):

In this manuscript Richter and colleagues use demethylase-thermostable group II intron RT tRNA sequencing (DM-tRNA-seq), one of a number of new tRNA deep sequencing approaches developed in recent years, to examine mitochondrial tRNAs in healthy and MERRF cells. The levels and m1A modification of the mutated tRNA-Lys in the MERRF cells were reduced, building on previous studies where modification of the wobble anticodon position of this tRNA was also found to be reduced. However restoration of the m1A modification by overexpression of the modifying enzyme TRMT61B revealed that it was not as important to the pathogenic effects on translation as the previously identified reduction in wobble modification of the tRNA. The work is well written and presented, however there is no real novelty and there are a few technical soundness issues that should be addressed in future submissions. Furthermore, the work is not as comprehensive or groundbreaking as typical studies published in this level of journal.

(1) Degree of novelty:

The authors should avoid hyperbole in their descriptions of their results: "unprecedented insight" is a bit over the top. Numerous studies have quantified mitochondrial tRNA abundance by deep

sequencing and even more comprehensively annotated their modifications, so even with the use of DM-tRNA-Seq, this work cannot be seen as “unprecedented” or new. As it stands the manuscript is a little shallow.

(2) Technical issues:

(a) For the deep sequencing analyses, using a.u. for tRNA abundance is not very useful. What about RPM, TPM etc.?

(b) The codon usage comparisons in Fig. 1E and 1G are not very meaningful when you consider the low complexity of the mitochondrial genome. There are only 11 mRNAs being translated in mitochondria and their levels can vary by an order of magnitude (e.g. see Chujo et al. NAR 2012), shifting the effective “codon concentration” dramatically. The authors should quantify the molar amounts of the 11 mRNAs in their same samples by Q-PCR or RNA-Seq and scale the codon abundances accordingly. This could then provide a meaningful comparison and a conceptual advance beyond what has been done before.

(c) The examination of oligo(A) addition to tRNAs is interesting, however to make their conclusions more compelling and increase the depth of the work the authors should provide these analyses for all 22 tRNAs. What if tRNA-LeuUUR is not representative of all the other tRNAs?

(d) For the restoration experiments the authors should provide western blots to verify MTO1, MTU1, and TRMT61B overexpression. This would help interpret the rescue of protein synthesis observed and if the lack of an effect from MTU1 overexpression is due to biological and not technical reasons.

Reviewer #1

1. Firstly, Chujo and Suzuki reported on 2012 (Chujo and Suzuki (2012) RNA; 18(12): 2269–2276) a partial m1A modification of tRNA^{Lys} position 58, which interacts with position 54 at the T-loop. They did not observe any Wild Type modification at position 54 of mitochondrial tRNA^{Lys}, but rather at position 58 (Figure 3C in Chujo and Suzuki, 2012). Is it possible that the authors of the current manuscript observed the previously described m1A at position 58 of tRNA^{Lys}? Position 54, which is mutated in MERRF (human mtDNA position 8344), and was claimed to undergo an m1A modification in wild type muscle cells, was never shown to undergo an m1A modification by any other group who studied the m1A modification of this tRNA. I find this very strange, especially, since the authors chose to totally ignore this discrepancy throughout the manuscript even while citing the 2012 paper by Chujo and Suzuki. This issue has to be carefully attended by the authors prior to further consideration of the manuscript.

We apologize for the confusion with numbering the m1A modification. Our data refers to the same m1A modification on mitochondrial tRNA^{Lys} reported by Chujo and Suzuki 2012 (RNA 18:2269) at position 58. In the manuscript, we listed the actual nucleotide position of the modification in the mitochondrial tRNA^{Lys}, which is 54 corresponding to m.8348 in the mitochondrial genome, because mitochondrial tRNAs do not have classical D and T loops. We thought Figures 1A and 4A illustrated the position of the m1A in relationship to the pathogenic mutation m.8344 A>G. Obviously, our decision led to confusion. Therefore, to be consistent with the previous report, we amended the manuscript accordingly to indicate the m1A occurs on position 58.

As the reviewer points out (comment #3), the m1A modification at position 58 forms a reverse-Hoogsteen base pairing with 5-methyluridine at position 54 in the T-loop. This interaction is seen in the mitochondrial tRNA^{Leu(UUR)}, where greater than 90% of A58 has the N¹-methyladenosine modification (see our data in Figure 2C, which matches the primer extension data published in Chujo and Suzuki 2012 (RNA 18:2269). However, in the tRNA^{Lys} there is an adenosine at position 54, which might account for the reduced level of the m1A58 in tRNA^{Lys}. In the discussion, we elaborate on the significance of the m1A58 to protein synthesis and the regulation of this modification, and a possible role in the molecular pathogenesis of MERRF.

2. In lines 143-144 the authors wrote: "DM-tRNA-seq allowed us to assess quantitatively differences in specific RNA modifications across all mitochondrial tRNAs between wild-type and MERRF-mutant cells". This method (Zheng et al Nature Methods 2015) is only relatively quantitative, while absolute quantification seems challenging. In the current manuscript the authors calculated modification index, which is OK, although again reflects relative abundance of the modifications, rather than absolute numbers. This should be clarified to avoid over interpretation of the data.

The manuscript has been modified to indicate that DM-tRNA-seq is only relatively quantitative. We would like to draw the attention of the reviewer to quantification of the m1A in tRNA^{Lys} with DM-tRNA-Seq was consistent with our primer extension results (Figure 4 B and C), and that reported by Chujo and Suzuki 2012 (RNA 18:2269), suggesting the quantification of RNA methyl modifications with our sequencing method is robust.

3. In line 176 the authors wrote a section entitled "m1A54 tRNA^{Lys} is linked to the MERRF pathogenesis in human skeletal muscle". This relates to major comment #1: Since previously an m1A modification was reported in tRNA^{Lys} position 58, and since this m1A at position 58 forms a reverse-Hoogsteen base-pairing with 5- methyluridine (m5U) at position 54 in the T-loop of tRNAs, it is very conceivable that the MERRF mutation affects such interaction, and in turn, affects the modification's level. I feel that this is what the authors observed in MERRF cells. Additionally, since the MERRF mutation is a point mutation, it is not surprising that other mtDNA recognition sites of TRMT61B are not affected, thus explaining why the levels of other m1A modifications throughout the mtDNA are unchanged. This interpretation should be recognized by the authors and the text throughout the manuscript should be altered accordingly.

Please see our reply to point #1 where this comment is addressed.

4. In lines 249-251 the authors wrote: "MTU1 catalyzes the thiol modification of the anticodon wobble base, a reaction that is required for human health (23), but appears to have no effect on mitochondrial protein synthesis in human fibroblasts (24)". The authors nicely cite differential effect of MTU1 driven thiol modification, but do not refer to tissue differences in mitochondrial m1A modifications, which are shown in Safra et al 2017 (they cited this paper elsewhere in the manuscript). As far as I am concerned there could be tissue differences in the level of the m1A modifications, which should be discussed in the current manuscript while analyzing only muscle cells.

We completely agree with the reviewer that there could be differences in the level of the m1A58 modification across tissues. One of the biggest challenges of human mitochondrial disorders is to understand the molecular basis for the exceptional tissue-specificity of these diseases that cannot be accounted by simply a biochemical defect to oxidative phosphorylation. We would like to point out that the m1A58 proportion in tRNA^{Lys} myoblasts matches that found in HeLa cells (Chujo and Suzuki 2012, *RNA* 18:2269), two dividing cell types of very different origins, and in post-mitotic skeletal muscle. Nonetheless, the differences in TRMT61B and ALKBH1 abundance and regulation across different cell types and tissues could have a profound impact on the level of this RNA modification. We now include a discussion of this point in the manuscript.

Minor comments.

1. In lines 120-122 the authors wrote: "Within the mitochondrial genome, the genes encoding these two tRNAs flank the 12S rRNA gene, and are transcribed together, along with the 16S rRNA, as a short polycistronic RNA at a level 25-fold higher than those of genes downstream along the H-strand (14)". The authors cited the 1993 paper by King and Attardi (ref 14 - *J. Biol. Chem.* 268, 10228–10237). A more recent report (Blumberg et al (2017) *Genome Res*;27(3):362-373) showed by using a quantitative in vivo nascent RNA transcription detection assay that there were tremendous quantitative differences in the transcription ratio between the light and heavy strands in various human cell lines, including HeLa cells – very different numbers than reported in the King and Attardi paper. The authors are asked to take into account the reported levels of transcription in this latter report (Blumberg et al. 2017) and change their statements accordingly.

We thank the reviewer for drawing our attention to the work of Blumberg et al. 2017 (*Genome Res* 27:362) as we inadvertently missed this publication. The findings of that paper were very illuminating and helpful to us, and have now been incorporated within our manuscript.

2. In lines 238-242 the authors wrote: "The m1A modification at position 947 of the 16S rRNA was proposed to be important for human mitochondrial ribosome subunit stability (20). Our analysis of the stability of mitochondrial ribosomal proteins from the small and large subunit indicated no difference following TRMT61B overexpression (Figure 4F)". I went and read citation 20 (Bar-Yaacov et al. *PLoS Biology*, 2016), and this paper suggested that the modification is presumably important to stabilize the structure of the mito-ribosome, not the levels of the mitochondrial ribosomal proteins. Thus, although a nice finding, I do not see any discrepancy between the findings of the current manuscript and citation 20. This should be acknowledged.

We were not trying to suggest a discrepancy between our findings and those of Bar-Yaacov et al. 2016 (*PLoS Biology* 14:e10002557). The m1A947 interactions occur within the mitochondrial 39S subunit and were proposed to stabilize ribosomal subunits, although never formally tested. Absence of this modification in bacterial ribosomes impaired translation and cell fitness. There is a tight interaction between mitochondrial ribosomal proteins and rRNA so that one is not stable without the other. Therefore, alterations to the stability of the large ribosomal subunit will be detectable by proportional differences in the steady-state levels of the large mitochondrial ribosomal proteins or the 16S rRNA. The point we were trying to make was that overexpressing the methyl transferase TRMT61B does not appear to produce a dominant-negative effect on the assembly of the mitochondrial ribosomal subunits. We modified the text to clarify this misunderstanding.

Reviewer #2

1. The title of the manuscript does not accurately reflect its content; it is not about the modification landscape, but is entirely about MERRF. Similarly, the abstract inadvertently implies that m1A58 of tRNA^{Lys} is important for stability of proteins, by the phrasing "By restoring the modification on the mitochondrial tRNA^{Lys}, we demonstrated the importance of the m1A54 to mitochondrial protein synthesis and nascent chain stability."

The reviewer is correct, the manuscript concentrates on the loss of a m1A modification on the tRNA^{Lys} m.8344A>G associated with the human MERRF syndrome, therefore, we changed the title to better reflect this point. The new title is "RNA modification landscape of the human mitochondrial tRNA^{Lys} regulates protein synthesis".

As for mitochondrial nascent chain stability from m1A58 modification, our data clearly shows the stability of MT-ATP6 is enhanced with this RNA modification when the steady-state level of the protein is assessed by SDS-PAGE immunoblotting (Figure 4F in the original submission and now Figure 5 in the revision). As requested in point #7, we now included an additional time point in the chase for metabolic labelling clearly illustrating a differential stability of mitochondrial nascent chains (Figure 5C). The basis for this differential effect is not known at this time but could be due to a number of factors, including the fact that these proteins need to assemble into large oligomeric complexes by different assembly pathways and in some cases also requiring co-factors (such as heme and copper moieties).

2. The data in Fig. 1B is mischaracterized, in the opinion of this reviewer. It seems clear that mitochondrial translation is almost completely defective in the MERRF cells, and the aberrant translation products are a minor but interesting side issue. The argument in the text lines 68-74 seems unfounded by the data presented, since translation looks defective, as expected. Moreover, there is no set of controls to validate that the aberrant translation in Fig. 1B is from the mitochondrial proteins. In this figure the red and black stars are not explained.

Mitochondrial translation is almost completely defective in the MERRF cells, however, a hallmark of the ³⁵S metabolic labelling pattern for this genetic mutation is the presence of aberrant translation products, one of which is called pMERRF: a premature translation product consistently found by many independent groups (see Boulet et al. 1992, Am. J. Hum. Genet 51:1187; and Enriquez et al. 1995, Nat Genet 10:47). These translation products can only be generated by mitochondrial protein synthesis because the assay uses anisomycin an effective inhibitor of cytosolic protein synthesis. Importantly, the aberrant sized fragments are no longer generated by restoring the m1A58 modification on tRNA^{Lys} in the MERRF cells (Figure 5A). To make the data more accessible to the general reader, we modified Figure 1 and Figure 5 and included more details on the mitochondrial translation assay.

2. The data in Fig. 1E and 1F seems ancillary to this reviewer, and if included, should contain a comparison to other known analyses for cytoplasmic tRNAs in other organisms to make a point. Moreover, in this reviewer's opinion the comparison to genomic codon usage is not nearly as relevant as that to highly expressed proteins, to mRNA levels, or some sort of codon adaptation index. In addition a trend line and R value or R-squared value would add credence to the claim of lack of correlation.

We agree with the reviewer, the data in Figure 1E and 1G is ancillary to the manuscript. Therefore, the data was moved to the supplementals (Figure S1B and S1C). An R-squared value has been added for both figures and shows no correlation ($r^2=0.0089$ and $r^2=0.0035$). For all intents and purposes mitochondrial gene expression is a closed system that is independent of the expression of the nuclear-encoded subunits of the oxidative phosphorylation complexes or other mitochondrial proteins. Therefore, comparing the codon usage of the 13 mitochondrially-encoded proteins, which are all hydrophobic membrane proteins, relative to a codon adaptation

index based upon highly expressed genes in the nucleus would not be an appropriate comparison. Transcription of mitochondrial DNA generates two long polycistronic messages that contain mRNAs and tRNAs, and are liberated as individual RNAs following processing by RNaseP and RNaseZ at the 5' and 3' ends, respectively. Despite the synthesis of the polycistronic message generating equal stoichiometry of the tRNAs per coding strand, our data shows differential stability of the mitochondrial tRNAs and variation between cell types. Moreover, the paper by Blumberg et al. 2017 (Genome Res 27:362) indicates a strand bias in the synthesis of nascent mitochondrial RNA synthesis that is cell type specific. This point has been mentioned now within the manuscript and reinforces the requirement for a cell type specific understanding of mitochondrial transcription. Obtaining a complete and thorough data set for the steady state levels of mitochondrial mRNAs is very much needed but is beyond the scope and focus of our manuscript.

4. The authors should define the wobble mutation in tRNALys by name (5-aurinomethyl-2-thiouridine), and should explain what heavy strand and light strand mean to readers not well versed in mitochondrial biology.

We followed the reviewer's suggestion to define the wobble base modification by name and have now included a schematic to illustrate the heavy and light strand of the mitochondrial genome to provide greater clarity to the reader (Figure 1A).

5. The data in Fig. 4B,E, and F is somewhat confusing. Fig. 4B and 4F appear to be from the same experiment or set of experiments, based on the legend, but 4F and 4E are from the same experiment based on the text.

6. It would be very useful to see a Western blot for the data in Fig. 4B to more accurately quantify or visualize translation provoked by overexpression of TRMT61B and MTO1 (instead of, or in addition to, Fig. 4C)

In our original submission, a Western blot was included to show the overexpression of TRMT61B and MTO1 (original Figure 4F, revised Figure 5E). We apologize that our presentation of the data led to confusion, therefore, we have separated the data into two figures (now Figure 4 and 5) and used better labelling.

7. It would be important to have a more complete time course of the instability of the proteins in Fig. 4E,F, accompanied by a comparison to the doubling time of the cells. This could help determine if and to what extent increased m1A (or 5-aurinomethylthiouridine) on the tRNA (and mRNA) impacts protein half-life relative to controls.

We followed the reviewer's suggestion and now have included an additional time point, which better illustrates the stability of the mitochondrial nascent chains (Figure 5C), and quantification of the data.

8. I am not sure why this manuscript shows dynamic regulation of modifications (line 305). It is true that modifications are dynamic, but the evidence here that they are is not that obvious.

We have amended the manuscript according to the reviewer's suggestion.

9. The discussion may need some focusing on the possibility that translation rates impact folding and stability of proteins.

We have now elaborated on the role of translation rates, folding and stability of mitochondrial nascent chains in the discussion.

Reviewer #3

(1) Degree of novelty:

The authors should avoid hyperbole in their descriptions of their results: “unprecedented insight” is a bit over the top. Numerous studies have quantified mitochondrial tRNA abundance by deep sequencing and even more comprehensively annotated their modifications, so even with the use of DM-tRNA-Seq, this work cannot be seen as “unprecedented” or new. As it stands the manuscript is a little shallow.

We respectively disagree with the reviewer that the field has already generated comprehensive studies on mitochondrial tRNA abundance by deep sequencing. Although sequencing results on mitochondrial tRNAs have been reported many times (e.g. Mercer et al. 2011, Cell 146:645-58), the field has not dealt with the fundamental problem of modification-interference with sequencing results. It is well established that Watson-Crick face methylations such as m1A severely interfere with quantitative analysis of RNA-seq (e.g. Hauenschild et al. 2015, Nucleic Acids Res. 43:9950-64). Therefore, the quantitative tRNA-seq results so far reported in the literature need to be considered with caution.

There are ample recent examples of method development to quantify tRNAs because of the modification-derived issues (e.g. Goodarzi et al. 2016, Cell 165:1416-1427; Gogakos et al. 2017, Cell Rep. 20:1463-1475). Importantly, the development of DM-tRNA-seq (Zheng et al. 2015 Nature Methods 12:835-837) was profiled by Nature Methods with a News and Views (Wilusz 2015, Nature Methods 12:821) highlighting the significance of the methodology to the field. As for modifications, the only comprehensive studies we know of are by mass spectrometry and none have profiled all 22 mitochondrial tRNAs in healthy and a disease state. The only full modification study by sequencing with validation was done by us using DM-tRNA-seq in cultured cells (Clark et al. 2016, RNA. 22:1771-1784). In this paper, we applied DM-tRNA-seq to study a disease-derived biological system for the first time (all previous DM-tRNA-seq papers were for method development). Furthermore, we applied tRNA-seq to study patient and human control tissues for the first time, and we produced new insights into the biology of mitochondrial diseases and tRNA modification dependent disease states. Perhaps “unprecedented” is a bit too strong to describe this study so we have modified the text to reflect a more restrained enthusiasm.

(2) Technical issues: (a) For the deep sequencing analyses, using a.u. for tRNA abundance is not very useful. What about RPM, TPM etc.?

We have now changed the units of the tRNA sequencing analysis to TPM (tRNA transcripts per million) for the figures.

(b) The codon usage comparisons in Fig. 1E and 1G are not very meaningful when you consider the low complexity of the mitochondrial genome. There are only 11 mRNAs being translated in mitochondria and their levels can vary by an order of magnitude (e.g. see Chujo et al. NAR 2012), shifting the effective "codon concentration" dramatically. The authors should quantify the molar amounts of the 11 mRNAs in their same samples by Q-PCR or RNA-Seq and scale the codon abundances accordingly. This could then provide a meaningful comparison and a conceptual advance beyond what has been done before.

As pointed out by reviewer #2, Figures 1E and 1G are ancillary to the main focus of the manuscript. Therefore, we moved the data to the supplementals.

We would like to point out that the analysis in Chujo et al 2012 (Nucleic Acids Res. 40:8033) for estimating the abundance of mitochondrial mRNAs has severe technical limitations that affect the interpretation of the data and preclude their use in our manuscript. These concerns are listed here.

(1). The individual reverse primers for generating the 11 cDNAs of the mitochondrial mRNAs are not consistently positioned along the transcripts, so in some cases primers anneal near the 5' end or in the middle of the mRNA whereas in others it is near the 3' end. Therefore, there is no evidence an accurate representation of the full-length mRNAs for all 11 transcripts was actually sampled and quantified. Moreover, the data set cannot control for any possible degradation (particularly 3'-5').

(2). The work from Li et al. 2017 (Mol. Cell 68:993) and Safra et al. 2017 (Nature 551:251) both show mitochondrial mRNA with m1A modification. Because of the RT enzyme used in Chujo et al 2012 (Nucleic Acids Res. 40:8033), the m1A modification will generate hard stops during reverse transcription introducing bias into the cDNA synthesis, affecting quantification of the mitochondrial mRNA stoichiometry.

Transcription of mitochondrial DNA generates two long polycistronic messages that contain mRNAs and tRNAs, and are liberated as individual RNAs following processing by RNaseP and RNaseZ at the 5' and 3' ends, respectively. Despite this equal stoichiometry of synthesis our data shows differential stability of the mitochondrial tRNAs that varies between cell types and is not reflected by codon usage. The research in Blumberg et al. 2017 (Genome Res 27:362) shows a strand bias for nascent RNA synthesis that appears to be cell type specific. Obtaining a complete picture of mitochondrial mRNA stoichiometry is a very important development but is tangential to the focus of our current manuscript, therefore, beyond the scope of our current study. We feel this is best left to a future study.

(c) The examination of oligo(A) addition to tRNAs is interesting, however to make their conclusions more compelling and increase the depth of the work the authors should provide these analyses for all 22 tRNAs. What if tRNA-LeuUUR is not representative of all the other tRNAs?

We thank the reviewer for the helpful suggestion because reanalysis of the sequencing data indeed detected differential oligo(A) addition across the mitochondrial tRNAs. This new analysis is now in Figure 3C.

(d) For the restoration experiments the authors should provide western blots to verify MTO1, MTU1, and TRMT61B overexpression. This would help interpret the rescue of protein synthesis observed and if the lack of an effect from MTU1 overexpression is due to biological and not technical reasons.

In our original submission, a Western blot was included to show the overexpression of TRMT61B and MTO1 (original Figure 4F, revised version Figure 5E). We apologize that our presentation of the data led to confusion, therefore, we have separated the data into two figures (now Figure 4 and 5) and used better labelling. As can be seen in the figure, there is no technical problems with our retroviral overexpression approach. As for MTU1, Sasarman et al. 2011 (Hum. Mol. Genet. 20:4634-4643) robustly demonstrated that loss of the thiol modification on the mitochondrial tRNA^{Lys} had no effect on the synthesis or stability of the mitochondrial proteins in dividing cells. Therefore, it was no surprise that no effect was observed with MTU1 overexpression. Mutations in MTU1 do lead to human disease but the molecular pathogenesis of the disorders apparently cannot be fully recapitulated biochemically in cultured cells.

Reviewers' comments:

Reviewer #1 (Remarks to the Author):

The authors adequately addressed my comments.

Reviewer #3 (Remarks to the Author):

In the revised manuscript the authors have made changes to the text and added another time point for the translation assays. However some of the key issues with the manuscript remain.

(1) Degree of novelty:

I'm not convinced that the work described here advances significantly on what Suzuki's group found, in terms of loss of modification of the MERRF tRNA-Lys and its effect on translation, over 10 years ago (e.g. Umeda et al. J Biol Chem 2005, Yasukawa et al. Mitochondrion 2002, Yasukawa et al. EMBO J, Yasukawa et al. FEBS Lett 2000). Furthermore, the m1A modification at position 58 has been documented in a number of studies (e.g. Chujo and Suzuki 2012) and in deep sequencing analyses (e.g. Li et al. Mol Cell 2017). The only advance here is that the m1A modification in the MERRF tRNA-Lys is shown to be lost. Therefore, the study doesn't provide the kind of substantial advance that would be expected for publication in this level of journal. An important point to note is that the overexpression of TRMT61B and consequent restoration of m1A modification at position 58 didn't actually rescue translation (apart from a slight recovery of MT-ATP6), while MTO1 overexpression and restoration of the lost mcm5s2U modification at position 34 did rescue translation. Therefore, this only proves that the loss of the mcm5s2U modification underlies of functional consequences of the 8344 MERRF mutation, as previously shown in detail by Suzuki's group. The loss of m1A at position 58 is only tangential to the pathology and its importance is overstated by the authors.

(2) Technical issues:

(a) The codon usage comparisons in Fig. 1E and 1G were shifted to the Supplementary Material, however they still suffer from the issues raised by both Reviewers #2 and #3 in the first round of review and should be removed if the authors cannot make the comparisons in a meaningful way.
(b) A western blot documenting correct overexpression has still not been provided for the restoration experiments with MTU1. Although it is true that previous studies have shown that loss of the modification introduced by MTU1 had no major effect in cultured cells, this doesn't mean that MTU1 might not have an effect on the MERRF tRNA when overexpressed, if this was a certainty why did the authors perform this experiment in the first place?

Reviewer Response Letter

Reviewer#3

1) Degree of novelty:

I'm not convinced that the work described here advances significantly on what Suzuki's group found, in terms of loss of modification of the MERRF tRNA-Lys and its effect on translation, over 10 years ago (e.g. Umeda et al. J Biol Chem 2005, Yasukawa et al. Mitochondrion 2002, Yasukawa et al. EMBO J, Yasukawa et al. FEBS Lett 2000). Furthermore, the m1A modification at position 58 has been documented in a number of studies (e.g. Chujo and Suzuki 2012) and in deep sequencing analyses (e.g. Li et al. Mol Cell 2017). The only advance here is that the m1A modification in the MERRF tRNA-Lys is shown to be lost. Therefore, the study doesn't provide the kind of substantial advance that would be expected for publication in this level of journal. An important point to note is that the overexpression of TRMT61B and consequent restoration of m1A modification at position 58 didn't actually rescue translation (apart from a slight recovery of MT-ATP6), while MTO1 overexpression and restoration of the lost mcm5s2U modification at position 34 did rescue translation. Therefore, this only proves that the loss of the mcm5s2U modification underlies of functional consequences of the 8344 MERRF mutation, as previously shown in detail by Suzuki's group. The loss of m1A at position 58 is only tangential to the pathology and its importance is overstated by the authors.

We respectfully disagree with Reviewer #3, however, on the relevance and importance of the m1A58 modification in the mitochondrial tRNA^{Lys} to mitochondrial protein synthesis and the pathogenesis of the MERRF syndrome. There appears to be confusion between the assays presented in Figure 5 and thereby the reviewer overlooks the importance of the m1A58 modification to mitochondrial protein synthesis. In this figure, the mechanism of mitochondrial translation elongation can only be studied by the incorporation of amino acids into the nascent chains during a pulse of ³⁵S metabolic labelling. Using this assay, restoration of the m1A58 modification on the tRNA^{Lys} completely rescued the rate of MT-ATP6 synthesis to wild type levels (Figure 5A and 5B) and eliminated the generation of aberrantly sized mitochondrial polypeptides characteristic of this MERRF mutation (Figure 5A). The reviewer appears to have overlooked this data and instead examined the steady state level of MT-ATP6 by SDS-PAGE immunoblotting (Figure 5D) to make the claim: "... consequent restoration of m1A modification at position 58 didn't actually rescue translation (apart from a slight recovery of MT-ATP6) ...". The steady state level of a protein is not indicative of the rate of synthesis. Moreover, our data clearly show that MTO1 overexpression alone cannot rescue the defect in mitochondrial nascent chains stability, showing that the mcm5s2U modification at position 34 is not the sole molecular determinant in the pathology of the MERRF syndrome and suggests other factors are important. This is why we feel it is important for the manuscript to highlight the missing m1A58 tRNA^{Lys} modification to the MERRF pathology because our data clearly demonstrated a robust role for this methyl modification in mitochondrial protein synthesis and nascent chain stability. A feature recognized by Reviewers# 1 and 2 in our original review.

(2) Technical issues:

(a) The codon usage comparisons in Fig. 1E and 1G were shifted to the Supplementary Material, however they still suffer from the issues raised by both Reviewers #2 and #3 in the first round of review and should be removed if the authors cannot make the comparisons in a meaningful way.

The figures on codon usage comparisons were removed from Figure S1 and the associated text deleted from the manuscript.

Reviewer Response Letter

(b) A western blot documenting correct overexpression has still not been provided for the restoration experiments with MTU1. Although it is true that previous studies have shown that loss of the modification introduced by MTU1 had no major effect in cultured cells, this doesn't mean that MTU1 might not have an effect on the MERFF tRNA when overexpressed, if this was a certainty why did the authors perform this experiment in the first place?

We tried to validate the MTU1 overexpression by commercially available antibodies but these reagents show no specificity to accurately determine the level of protein overexpression. Unfortunately, this is a common problem with commercial antibodies. To circumvent this issue, we used a semi-quantitative RT-PCR assay to show *MTU1* mRNA overexpression by retrovirus (Figure S2E). Here, we used a gene-specific primer in reverse transcription for the cDNA synthesis followed by PCR with different cycle numbers. This data shows robust overexpression of *MTU1* with our retroviral system. In contrast, there was no difference in the abundance of *ATAD3A* mRNA, a nuclear-encoded mitochondrial protein.